# Forecasting the Monkeypox Outbreak Using ARIMA, Prophet, NeuralProphet, and LSTM Models in the United States

Bowen Long *, Fangya Tan and Mark Newman

Department of Analytics, Harrisburg University of Science and Technology, Harrisburg, PA 17101, USA
* Correspondence: blong@my.harrisburgu.edu

**Abstract:** Since May 2022, over 64,000 Monkeypox cases have been confirmed globally up until September 2022. The United States leads the world in cases, with over 25,000 cases nationally. This recent escalation of the Monkeypox outbreak has become a severe and urgent worldwide public health concern. We aimed to develop an efficient forecasting tool that allows health experts to implement effective prevention policies for Monkeypox and shed light on the case development of diseases that share similar characteristics to Monkeypox. This research utilized five machine learning models, namely, ARIMA, LSTM, Prophet, NeuralProphet, and a stacking model, on the Monkeypox datasets from the CDC official website to forecast the next 7-day trend of Monkeypox cases in the United States. The result showed that NeuralProphet achieved the most optimal performance with a RMSE of 49.27 and $R^2$ of 0.76. Further, the final trained NeuralProphet was employed to forecast seven days of out-of-sample cases. On the basis of cases, our model demonstrated 95% accuracy.

**Keywords:** Monkeypox; forecasting; ARIMA; LSTM; Prophet; NeuralProphet





## 1. Introduction

Monkeypox is a zoonotic disease, a virus transmitted initially from animals, belonging to the family of smallpox [1]. In 1980, the WHO declared Monkeypox the most significant orthopoxviral in public health [1]. Historically, the overall fatality ratio caused by Monkeypox can reach 11%, with a higher death ratio among children [2]. The USA has seen a dramatic increase in the occurrence of Monkeypox since May 2022, as cases have been reported from multiple non-endemic countries. On 23 July 2022, the WHO declared Monkeypox a global public emergency [3]. As of September 2022, the United States accounted for approximately 40% of global cases. In the USA, over 99% of Monkeypox patients are men, 94% of whom reported having men-to-men sex [4]. Therefore, social media was flooded with homophobic discussions. Gonsalves pointed out the similarity between HIV and Monkeypox regarding domestic and social responsibility, which led to the stigmatizing of LGBTQ groups and other social issues in the public health crisis [5]. Several researchers on smallpox historical data have indicated that smallpox transmission is greatly enhanced by dry weather [6,7]. Additionally, Grais concluded that smallpox cases would increase by 11–30% by air travel, depending on the condition of the initial city [8]. Being a variant of the smallpox family, we expect a similar pattern of increasing cases with the arrival of the dry season and the gradual lifting of COVID-19 air travel restrictions. A recent paper published in *Nature Medicine* showed that the current virus had accelerated evolution by mutating far more frequently than expected [9]. Moreover, experts warned about the improved ability of the Monkeypox virus to infect humans [10]. Therefore, compounded with the virus mutation, the epidemic will likely have a second outbreak, and we hope to fight the Monkeypox virus together as a community.

By comparing the performance of different models in this research, we proposed the most robust and efficient model for forecasting the U.S. Monkeypox trend over the next seven days. Our model is expected to alert health experts in the USA before the crisis and

help them implement effective prevention policies such as allocating vaccines or resources. The existing research literature on Monkeypox prediction is sparse, and some predictions based on early stage patterns either underestimate or overestimate the epidemic. For example, McAndrew predicted that there is only a 25% probability that the WHO will declare the Monkeypox outbreak as a public health crisis of international concern in May this year [11]. However, in July this year, the CDC declared Monkeypox a global public emergency. Majumder forecasted the prevalence of Monkeypox cases over 100 days using a polynomial-based network and showed that the number of cases would continuously increase from 28 July 2022, to 20 October 2022 [12]. Nevertheless, since August 2022, the primary affected countries of the epidemic, such as the United States and Spain, have shown a downward trend to varying degrees [13]. Nevertheless, there is still the possibility of a second epidemic outbreak with the upcoming dry season, virus mutations, school returns [14], and travel restrictions lifting.

Historically, the ARIMA (auto regressive integrated moving average) model has been the most popular forecast model in the financial industry and market demand [15]. A number of studies have applied ARIMA in disease prediction, particularly with data following a SIR (susceptible–infectious–recovered) model, such as Ebola [16], SARS [17], HIV [18–21], and COVID-19 [22–26]. Spath's paper showed that the Monkeypox outbreak in 2022 data followed the modified SIR model distribution [2,27]. Sharing SIR features, we decided to use ARIMA as one of our models to predict Monkeypox cases. Due to ARMA's linear nature and reliance on a stationarity assumption, it may be challenging to achieve remarkable performance considering Monkeypox's non-stationary feature, which was identified in our research data.

The continued development of deep learning in various fields has led to ample evidence that LSTM (long short-term memory) is one of the most powerful algorithms that can predict the future from sequences of variable lengths for both linear and non-linear data [28]. This feature overcomes the barrier of ARMA's linearity and stationary data assumption. Hence, researchers have shown a keen interest in this innovative and dynamic time-series forecasting approach in public-health-related fields, such as general disease classification [29], cardiovascular health projection [30,31], and infectious disease prediction [21]. Chimmula and Rauf reported over 90% accuracy when utilizing LSTM to estimate COVID-19 numbers in their studies [32,33]. Besides the pervasive application of LSTM in COVID-19 predictions, it is also a desired model to predict Zoonotic diseases such as Chickenpox [34]. Essien's study indicated an RMSE of 92.57 when involving the LSTM method in their model to forecast Chickenpox. Moreover, Zhu manipulated LSTM in sexually transmitted disease case prediction such as in syphilis, AIDS, and gonorrhea in China and concluded that LSTM is a robust predictive model that can be applied for STD control and prevention [21]. Moreover, LSTM and ARIMA models appear to be closely linked and often compared [32] in forecasting performance. The resemblance of Monkeypox as a zoonotic disease with Chickenpox; primary sexually transmitted diseases such as syphilis, AIDS, and gonorrhea; and a highly contagious public health emergency such as COVID-19 motivated us to choose LSTM as another model to predict Monkeypox cases. Compared with previous research, in addition to extensive hyperparameter tuning, we also tested different input vector sizes and adjusted them to fit the trend of Monkeypox best. The problem is that LSTM typically requires a large dataset and comprehensive hyperparameter tuning for outstanding performance. Otherwise, it might be overfitting. In contrast to COVID-19, Monkeypox case numbers are relatively low in terms of infection scope and shorter in data collection.

Prophet is a novel forecasting approach from Facebook in 2018; it performs best for datasets with strong seasonal effects [25]. Compared with LSTM, it requires less hyperparameter tuning and fewer data. Increasingly more researchers are applying Prophet in disease forecasting. For instance, Satrio's paper established the conclusion that the accuracy level of the Prophet (91%) model outperformed the ARIMA (<50%) model when predicting COVID-19 cases in Indonesia [25]. Moreover, Xie concluded that Prophet

outperforms ARIMA in predicting the incidence of hand, foot, and mouth disease, a kind of infectious disease among children, in Wuhan, China [35]. In addition, because our data are based on CDC daily reported cases, there are relatively few reported cases on Saturday and Sunday due to the closure of most medical institutions or short hours. This seasonality in the data due to reporting timing can be handled well by Prophet.

NeuralProphet [36] is an extension of the Prophet model that retains all the advantages of Prophet while combining the scalability of neural networks with the interpretability of AR models by introducing an improved backend based on PyTorch [37] and an autoregressive network called AR-Net to improve its accuracy and scalability. The breakthrough forecasting technique has been applied in multiple research domains and has achieved remarkable estimation results. For example, Velásquez used NeuralProphet to predict the energy of PV solar plants with a MAPE of 5.93%, which outperformed ARIMA-LSTM's MAPE of 10.57% [38]. Moreover, many researchers used NeuralProhpet in COVID-19 case predictions and found it to be the most accurate among other models [39,40]. Theoretically, NeuralProphet should always have equal or better performance than Prophet [36]; thus, we used NerualProphet as our fourth model.

In summary, we used the ARIMA, LSTM, Prophet, and NeuralProphet models to develop a robust univariate time series model to predict U.S. Monkeypox reported cases in the coming seven days. In addition, we proposed a stacking model of the four models as another alternative aiming for a well-balanced performance. Figure 1 shows the research framework employed in this research.

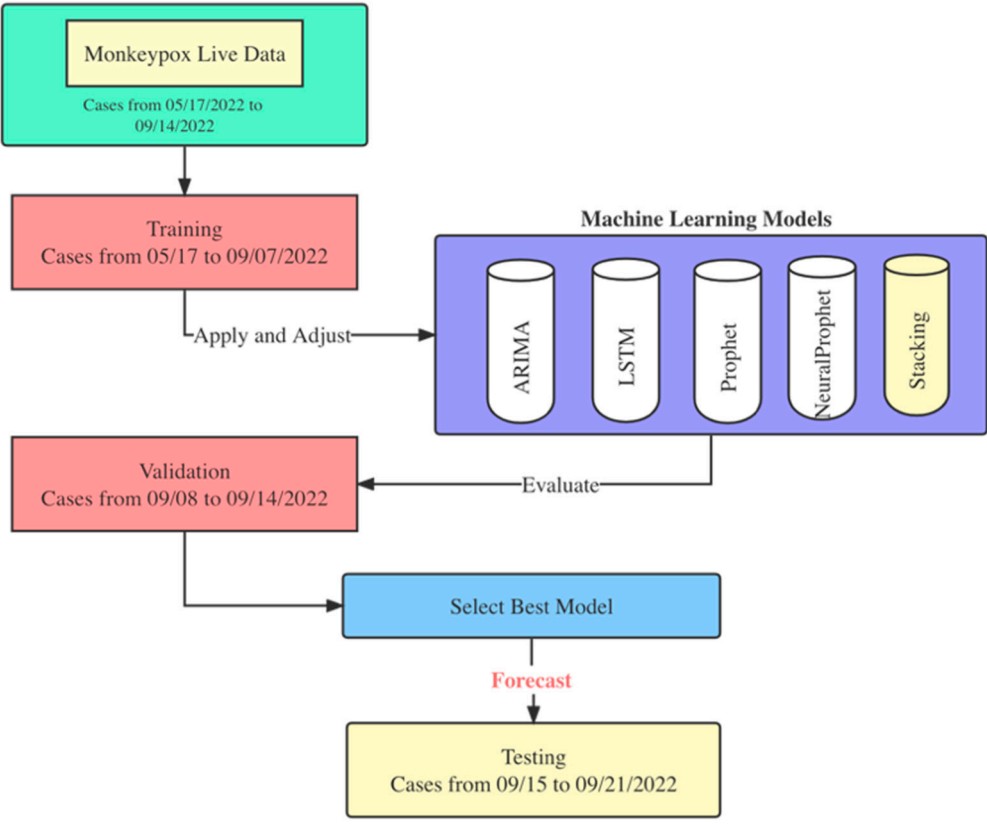

**Figure 1.** Object process diagram of Monkeypox comparative analysis.

## 2. Methods and Models

### 2.1. Dataset

The datasets used in this research were collected from the CDC's official website [41], being provided in time series format with the number of reported cases in the United States on each date from 17 May 2022. We split the datasets into a training set (reported cases from 17 May 2022 to 7 September 2022, as of 21 September 2022) on which our models were

trained. A validation set (reported cases from 8 September 2022 to 14 September 2022, as of 21 September 2022) was used to evaluate the model's performance and determine the best model for an out-of-sample forecast. Using the highest performing model selected on the validation set, we forecasted its performance on the testing set (reported cases from 15 September 2022 to 21 September 2022, as of 5 October 2022). Since there could be a delay for a contracted case to be submitted to CDC, the cases from 15 September to 21 September 2022, as of 21 September 2022, were neglected in this research to minimize the latency issues. Figure 2 summarizes the U.S. reported cases from 17 May 2022 to 14 September 2022, as of 21 September 2022.

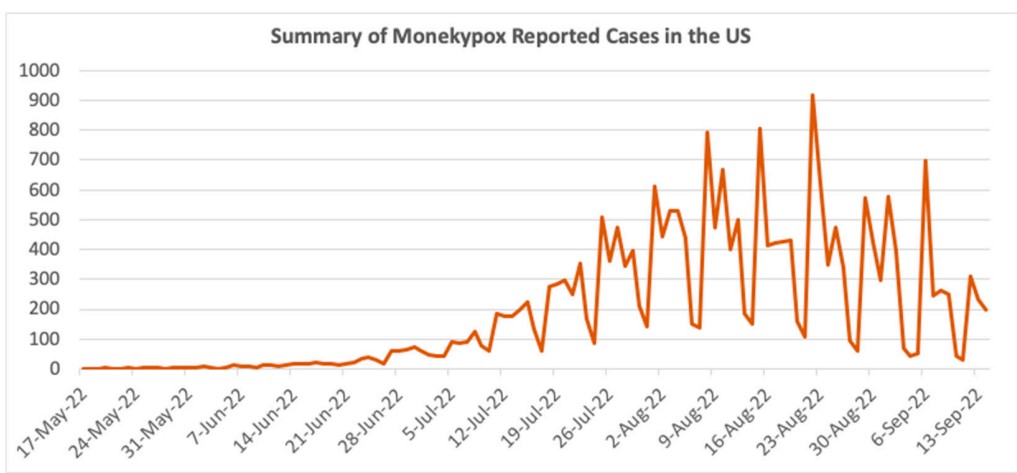

**Figure 2.** Summary of Monkeypox reported cases from 17 May 2022 to 14 September 2022 in the USA.

### 2.2. Forecasting with ARIMA

ARIMA (autoregressive integrated moving average), also known as ARIMA(p, d, q), is one of the most common statistical models used for time series forecasting—it predicts future values using past values and errors. The integrated I stands for the number of times differencing is needed to make the times series stationary.

The ARMA model requires stationary series with a constant mean, variance, and covariance. Figure 3 shows the decomposition of Monkeypox time series data, a strong trend, and a seasonal pattern observed. We implemented the Dickey–Fuller test to double-check the stationarity of the data. We obtained a p-value > 0.05. Hence we concluded that the series is not stationary.

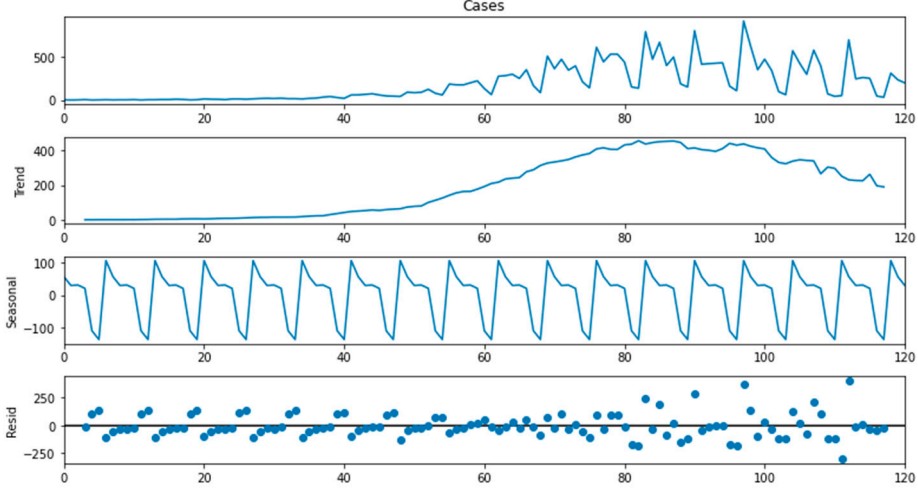

**Figure 3.** Decomposition of reported Monkeypox cases from 17 May 2022 to 14 September 2022 in the USA.

To identify order p, d, q. First, we compared the model performance without differencing terms, and with first-order differencing, the model outperformed; thus, we set the order of difference d = 0. Then, to identify order p and order q, we obtained the autocorrelation coefficient (ACF) and partial autocorrelation coefficient (PACF). Through the diagrams, we found that the PACF was cut after the first lag but there was no clear-cut for ACF, and therefore we set p = 1. To further identify order q, we tested it from 0 to 10. We found that the AIC and BIC values stayed stable with the increase in order q, and thus we set q = 0. Therefore, our ARIMA model can be defined as ARIMA (1,0,0), which is also a simple autoregressive model.

Since a strong seasonal and trend pattern was observed in the data, it was challenging for the ARIMA model to achieve remarkable performance. Therefore, in this research, we also utilized LSTM, Prophet, and NeuralProhet to forecast, which are not necessarily applied to stationary series.

### 2.3. Forecasting with LSTM

The LSTM (long short-term memory) neural network is one of the most advanced models to forecast time series, and it is an improved version of the traditional recurrent neural network (RNN). Hochreiter and Schmidhuber proposed the long short-term memory network [28]. Unlike the RNN, the LSTM adds a memory unit to judge whether the information is valuable, solving the vanishing gradient and gradient explosion problems during multi-sequence training. This improvement allows for better performance on longer sequences.

To train an LSTM network, we need to set a sequence using previous time steps as input and sequences in the next time steps as output. In this research, we set the timesteps for each input to 14 at the start. Each observation consisted of an input vector of reported cases from the past 14 days and an output vector of 7 for the reported cases in the next 7 days. After determining an optimal set of hyperparameters, we also tested the size of the input vector and selected an optimal one. For effective network learning, we standardized the series with a mean of 0 and a standard deviation of 1 before fitting the LSTM model.

LSTM is prone to overfitting on a small dataset. Moreover, it typically requires comprehensive hyperparameter tuning for it to perform well. Therefore, we also utilized Prophet and NeuralProphet in this research, requiring less hyperparameter tuning and less data.

### 2.4. Forecasting with Prophet and NeuralProphet

The Prophet is a time series prediction algorithm developed by the Facebook team. The algorithm combines time series decomposition and machine learning algorithms and can predict time series with missing values and outliers. It divides the time series into the following parts:

$$y\,(t) = g\,(t) + s\,(t) + h\,(t) + \alpha \tag{1}$$

In this equation, g(t) represents the trend item, which reflects the non-periodic change trend of the time series; s(t) represents the seasonal item, which reflects the cyclical changes of the time series, such as weekly and annual seasonal changes; h(t) represents the holiday item, which reflects the irregular changes that last from one day to several days such as holidays; the last $\alpha$ represents the residual item, which is normally distributed. Then, each item is fitted separately, and the final cumulative result is the prediction result of the Prophet algorithm.

NeuralProphet is an extension of the Prophet model that retains all the advantages of Prophet while combining the scalability of neural networks with the interpretability of AR models by introducing an improved backend based on PyTorch and an autoregressive network called AR-Net [42] to improve its accuracy and scalability. The authors of AR-Net propose the learning of an autoregressive forecasting model with a feedforward neural network instead of using a linear model fit with least squares [42]. As a result, a multi-layer feedforward neural network is more expressive than a linear model, and AR-Net scales

much better to larger datasets and inputs. Prophet and NeuralProphet perform the most effective for datasets with substantial seasonal effects.

Since our data are based on CDC daily reported cases, there are relatively few reported cases on Saturday and Sunday. This is due to the closure of most medical institutions or short hours. This seasonality in the data due to reporting timing can be handled well by Prophet. Figure 4 shows the Monkeypox reported cases by weekday according to our data.

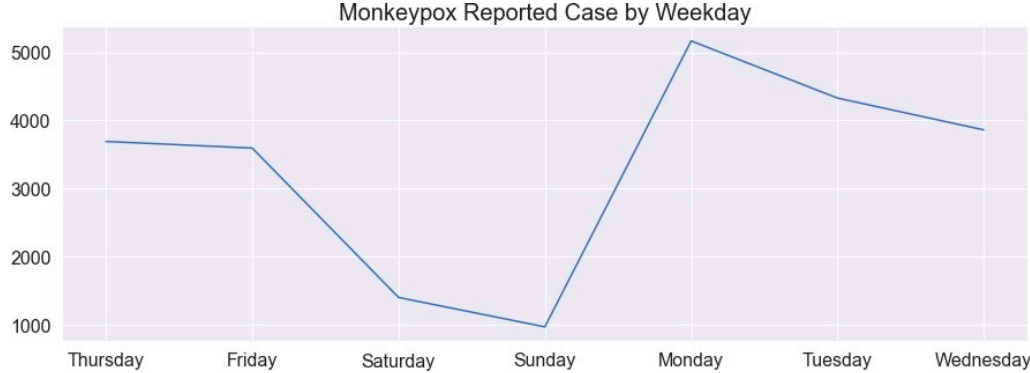

**Figure 4.** Reported Monkeypox cases by weekday.

*2.5. Forecasting with Stacking Model*

To achieve a well-balanced performance, we combined the outputs of ARIMA, LSTM, Prophet, and NeuralProphet models and built a stacking model. Model stacks can be made in many ways, such as averaging, weighted averaging, and a meta-learner trained using other machine learning models on individual learners. In this research, we used model averaging. Figure 5 shows the architecture of the stacking model.

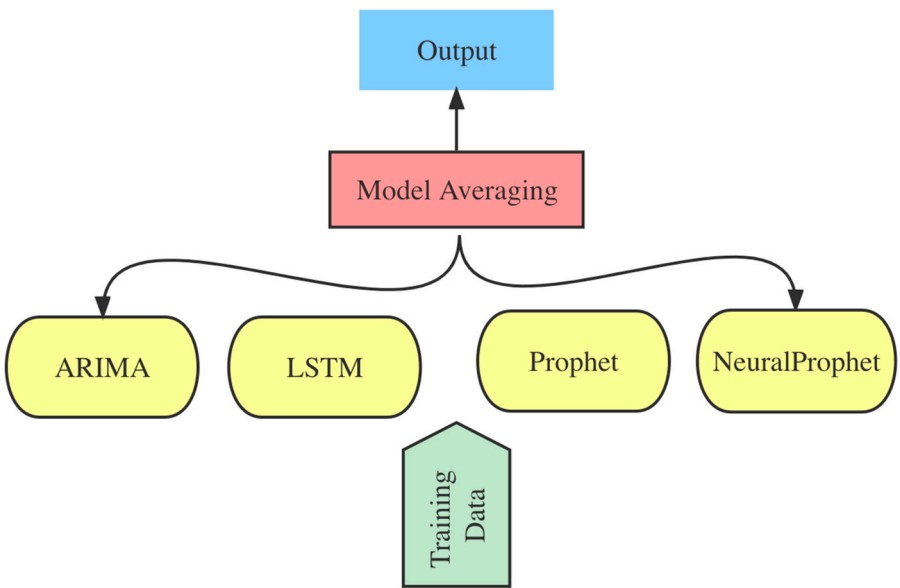

**Figure 5.** Architecture of the stacking model.

**3. Results and Discussion**

ARIMA, LSTM, Prophet, NeuralProphet, and stacking models were trained using the reported cases of Monkeypox between 17 May 2022 and 7 September 2022, as of 21 September 2022. Then, we evaluated the model performance on the validation set on the basis of reported cases between 8 September 2022 and 14 September 2022, as of 21 September 2022. After selecting the best model on the validation set, we used its forecast on the testing set, reported cases from 15 September to 21 September 2022, as of 5 October 2022.

For ARIMA, first, we found that the model without differencing terms outperformed the model with first-order differencing. Then, we determined the order p and q on the basis of ACF and PACF diagrams, and AIC and BIC values. As a result, our ARIMA model is defined as ARIMA (1,0,0), the same as a simple first-order autoregressive model. For the LSTM model, first, for each observation at a timestep, we set the size of the input vector to be 14 and the output vector to be 7. Then, we adjusted the number of hidden layers of the network, the number of output units of each hidden layer, the epochs of model training, the batch size, and the learning rate. After determining an optimal set of hyperparameters, we tested different input vector sizes on the training set to select the optimal size. With Prophet and NeuralProphet, we used the default parameters, which are automatically tuned. Lastly, we averaged the prediction results of ARIMA, LSTM, Prophet, and NeuralProphet models as the output of the stacking model.

The LSTM experiment results showed that RMSE remained stable when the learning rate was less than 0.1. However, when increased to 0.1, the error rate was too large, and thus we chose the learning rate at 0.02, which had the smallest RMSE on the training set. Regarding the number of epochs, the RMSE decreased with the increase in epochs and stayed stable after expanding to 50, and therefore we set epochs as 50. As for the number of hidden layers and output units of each layer, we observed that the RMSE increased when the structure of hidden layers became more complex, meaning more layers and more units. Hence, we chose one layer with 128 output units as our hidden layer structure, with the most straightforward system and lowest RMSE. As for batch size, it showed a similar pattern as the number of epochs. After increasing the batch size to 10, the RMSE no longer decreased, and thus we chose 10 for the model. After determining an optimal set of hyperparameters, we tested the different sizes of the input vector on the training set to select the optimal one. We observed that the RMSE showed an upward trend with the increase in input vector size. Therefore, to minimize RMSE, we chose an input vector size of 7.

Figure 6 compares the actual reported cases and the prediction results of four models on the validation set. There were 1332 reported cases of Monkeypox from 8 September 2022 to 14 September 2022, as of 21 September 2022. By comparing the results of the models, in addition to Prophet, we found that the prediction results of all the other four models showed an overall downward trend. The downward trend was consistent with the actual change in the case. Prophet and NeuralProphet are a superior fit for the daily dynamics of cases, probably due to the handling of the data seasonality resulting from reporting timing. However, LSTM and stacking were delayed by a day compared to the actual case peak, and ARIMA lacked the separation between daily movements and flattening. Table 1 compares the performances of four models using MAPE, MAE, RMSE, and R2. In terms of error rate, NeuralProphet had the lowest MAPE. On the other hand, Prophet and LSTM had higher MAPE among the four models, and in terms of MAE and RMSE, NeuralProphet also had the most accurate results, while stacking had average performance among all the models.

Moreover, LSTM and Prophet had negative $R^2$ values, which indicates that the residual sum of squares was larger than the total sum of squares, and therefore these two models fit worse than a horizontal line of cases. In conclusion, NeuralProphet performed the best among all models. We also used it to forecast the reported cases from 15 September 2022 to 21 September 2022, as shown in Figure 7.

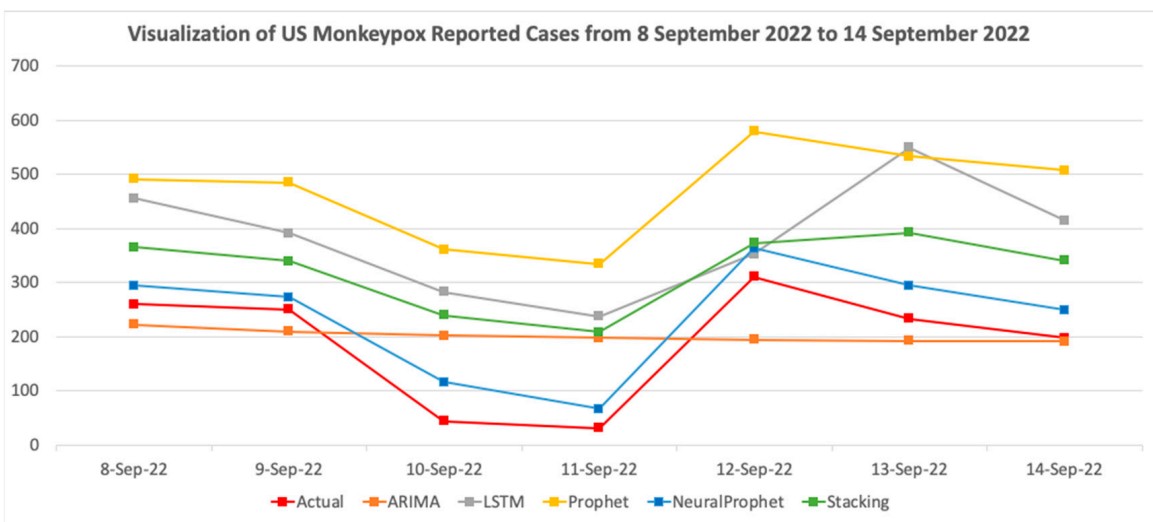

**Figure 6.** Comparison between actual and prediction for the next 7 days.

**Table 1.** Evaluation metrics for the 4 models.

| Metrics | ARIMA | LSTM | Prophet | NeuralProphet | Stacking |
|---------|-------|------|---------|---------------|----------|
| MAPE | 136% | 223% | 314% | 51% | 175% |
| MAE | 80.79 | 193.19 | 280.02 | 46.62 | 132.86 |
| RMSE | 100.38 | 208.58 | 282.02 | 49.27 | 140.34 |
| R2 | 0.01 | −3.27 | −6.80 | 0.76 | −0.93 |

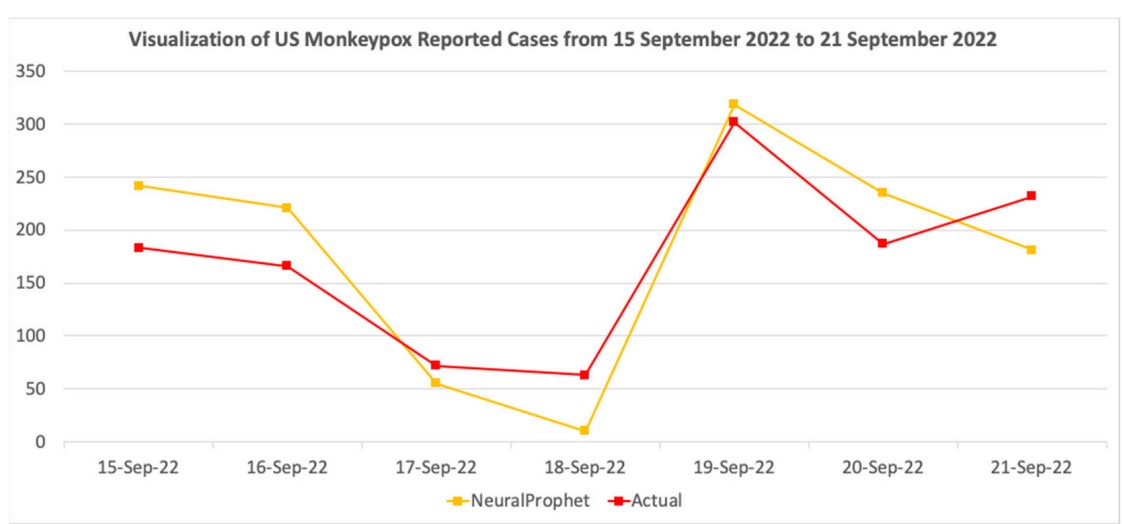

**Figure 7.** Forecast using NeuralProphet for the next 7 days from 15 September 2022 to 21 September 2022.

As shown in the figure above, using the final trained NeuralProphet model, we predicted there would be about 1263 reported cases from 15 September to 21 September 2022. Due to the closure of most medical testing facilities on weekends, 17 September and 18 September 2022 have the least reported cases. Overall, the cases were still on a downward trend. The rate of decline was roughly the same as that of the week of 8 September to 14 September 2022. On the basis of actual reported cases as of 5 October 2022, there were a total of 1205 reported cases. Therefore, our model showed 95% accuracy.

Further, we compared NeuralProphet prediction performance on the validation and testing sets. We found that with the decrease in overall reported cases, the performance

of NeuralProphet improved. Table 2 shows the result of the comparison, both of the two periods with positive $R^2$, with the latter period having lower error rates. Therefore, we estimated that with strong and effective policy measures, case development will be more stable, and NeuralProphet will be more capable of forecasting the trend.

**Table 2.** NeuralProphet comparison between validation and testing sets.

| NeuralProphet Prediction Period | 8–14 September 2022 | 15–21 September 2022 |
| --- | --- | --- |
| *Actual reported cases* | *1332* | *1205* |
| MAPE | 51% | 32% |
| MAE | 46.62 | 42.86 |
| RMSE | 49.27 | 45.98 |
| $R^2$ | 0.76 | 0.65 |

Several limitations of our study need to be highlighted. First, the data for our analysis were from CDC-reported cases. Asymptomatic persons with milder symptoms may have been missed in this case series because they did not seek medical care. Moreover, due to data latency issues, the delay for a contracted case to be reported could also bias the actual trend of case development on each day. Further, with the arrival of the dry season, the gradual lifting of COVID-19 travel restrictions, and the virus mutation, the epidemic is likely to have a second outbreak, and the performance of the model trained on the basis of the current period may decrease.

**4. Conclusions and Future Work**

Starting from better monitoring and epidemic prevention of Monkeypox, this research utilized machine learning models such as ARIMA, Prophet, NeuralProphet, LSTM, and a stacking model to predict reported cases of Monkeypox in the next seven days. The training and validation data came from reported cases on the CDC official website from 17 May 2022 to 14 September 2022, as of 21 September 2022. The testing data came from reported cases from 15 September 2022 to 21 September 2022, as of 5 October 2022. According to the validation set, NeuroProphet achieved the most accuracte prediction results, with 51% MAPE and 0.76 $R^2$ on 8 to 14 September 2022, as of 21 September 2022. At the same time, we used the final trained NeuralProphet model to forecast the reported cases on the week of 15 September. The forecast shows there would be about 1263 reported cases from 15 September to 21 September 2022. Compared with actual reported cases, our model achieved 95% accuracy. The cases generally showed a downward trend, and the decline rate was roughly the same as that of the week of 8 September to 14 September 2022. Judging from the data pattern and forecast results, the current anti-epidemic measures taken by the U.S. government are effective. However, it is still too early to conclude the long-term overall performance.

In summary, this is the first study using machine learning approaches to predict reported Monkeypox cases in the USA. The predicted results have positive guiding significance for supervising the epidemic and allocating vaccines. Furthermore, we discovered that Monkeypox, as a highly contagious zoonotic disease, not only follows the SIR epidemic distribution but also shows a sexually transmitted pattern and seasonal pattern, resulting from data collection methods. By leveraging the NeuralProphet model, we obtained a descent forecast for this disease. Therefore, we believe NeuralProphet has excellent potential to be applied to forecasting diseases that share characteristics similar to Monkeypox, such as Chickenpox, HIV, SARS, and COVID-19. As a result, health experts and policymakers can take informed actions as early as possible to better prepare the resources needed to fight against the disease. Moreover, currently, this study focused on univariate time series forecasting in the USA, but it can be extended to forecast the Monkeypox outbreak globally, and external variables that could accelerate a second outbreak, such as weather temperature and air passenger traffic, can be incorporated in order to improve the forecasting accuracy of the multivariate time series forecasting model.

**Author Contributions:** Conceptualization, B.L. and F.T.; methodology, B.L. and F.T.; software, B.L.; validation, B.L.; formal analysis, F.T.; investigation, F.T.; resources, M.N.; data curation, M.N.; writing—original draft preparation, B.L. and F.T.; writing—review and editing, F.T., M.N. and B.L.; visualization, B.L.; supervision, F.T.; project administration, M.N. All authors have read and agreed to the published version of the manuscript.

**Funding:** This research received no external funding.

**Data Availability Statement:** Publicly available datasets were analyzed in this study. These data can be found here: https://www.cdc.gov/poxvirus/monkeypox/response/2022/us-map.html (accessed on 5 October 2022).

**Conflicts of Interest:** The authors declare no conflict of interest.

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
