# Peer review of "Forecasting the Monkeypox Outbreak Using ARIMA, Prophet, NeuralProphet, and LSTM Models in the United States"

_forecasting, doi:10.3390/forecast5010005_

Round 1
Reviewer 1 Report
see attached pdf

Reviewer 2 Report
The paper compares four models for the prediction of daily data of Monkeypox cases. The models considered are well established in the literature and no alternative variation or modelling to the existing ones is proposed.
Although the analysis of Monkeypox case data is very pertinent and useful, the analysis methodologies are not innovative.
Even the selection and application of the 4 models is not well described. The estimated ARIMA model probably indicates a seasonal structure, although the authors did not even consider this hypothesis. Another evidence that the competitive models are not good options is the evidence of negative R2.
Reviewer 3 Report
In my opinion, the article is well balanced and does not require mandatory specific revisions. The structure is adequate, the methodology applied is correct and there are no structural inaccuracies or gaps. Optionally, the authors could deepen in the discussions a comparative assessment with other similar studies, analyzing different realities in different countries.
Round 2
Reviewer 2 Report
I think the analysis of this data is pertinent but the revised version of the paper maintains the previous approach of adopting established models and that, in general, it is not clear that these are the appropriate models given the trend and seasonality of the variable. The paper does not present the methodological innovation expected to be published in the Forecasting journal.